# Effect of Condom Use after CIN Treatment on Cervical HPV Biomarkers Positivity: Prolonged Follow Up Study

**DOI:** 10.3390/cancers14143530

**Published:** 2022-07-20

**Authors:** George Valasoulis, Georgios Michail, Abraham Pouliakis, Georgios Androutsopoulos, Ioannis. G. Panayiotides, Maria Kyrgiou, Alexandros Daponte, Evangelos Paraskevaidis

**Affiliations:** 1Department of Obstetrics & Gynaecology, University Hospital of Larisa, 41334 Larisa, Greece; daponte@med.uth.gr; 2Hellenic National Public Health Organization-ECDC, 15123 Athens, Greece; 3Department of Obstetrics & Gynaecology, University Hospital of Patras, 26504 Patras, Greece; gmichail@upatras.gr (G.M.); androutsopoulos@upatras.gr (G.A.); 4Second Department of Pathology, Attikon University Hospital, National and Kapodistrian University of Athens, 12462 Athens, Greece; apouliak@med.uoa.gr (A.P.); ioagpan@med.uoa.gr (I.G.P.); 5West London Gynaecological Cancer Centre, Imperial College Healthcare NHS Trust, London W12 0HS, UK; m.kyrgiou@imperial.ac.uk; 6Institute of Reproductive and Developmental Biology, Department of Metabolism, Digestion and Reproduction-Surgery and Cancer, Imperial College London, London W12 0NN, UK; 7Department of Obstetrics & Gynaecology, University Hospital of Ioannina, 45500 Ioannina, Greece; eparaske@uoi.gr

**Keywords:** condom, human papillomavirus, HPV, HPV DNA, mRNA E6 & E7, biomarkers, CIN, colposcopy, lifestyle

## Abstract

**Simple Summary:**

The definite effects of consistent condom use following cervical local excisional procedures on preventing recurrent preinvasive disease as well as in biomarkers of HPV expression have not been studied so far. We embarked on a prospective observational study enrolling over 200 individuals who received strong advice for consistent condom use after undergoing local surgical CIN treatment. We assessed in the mid and long term (six and twenty-four-month post-op intervals, respectively) rates of CIN recurrence and the expression of HPV dependent and other biomarkers (HPV DNA, HPV mRNA E6 & E7, p16) in correlation with consistency of use. A favorable effect of routine condom use in rates of CIN relapse and biomarker expression was evident at the 6-month follow up; this was more pronounced at the 2-year assessment. In conclusion, consistent condom use following cervical local excisional treatment might influence favorably rates of CIN recurrence and biomarkers of HPV expression.

**Abstract:**

Background: Several factors contribute in the cervical healing process following local surgical treatment; in a previous work our group has documented a beneficial mid-term role of regular condom use immediately postoperatively in terms of CIN relapse prevention and expression of active viral biomarkers. Materials and Methods: Aiming to investigate whether the favorable contribution of consistent condom use could be extrapolated in the longer term, we conducted a prospective single center observational study including women scheduled to undergo conservative excisional treatment for CIN (LLETZ procedure). In all women a strong recommendation for consistent use for the first 6 months was given. For 204 women who underwent the procedure and completed successfully the two-year follow up a complete dataset of HPV biomarkers’ results obtained six months and two years postoperatively was available. Patients were asked to complete a questionnaire to assess condom use compliance. A 90% compliance rate represented the threshold for consistent use. An LBC sample was obtained and tested for HPV genotyping, E6 & E7 mRNA by NASBA technique as well as flow cytometry, and p16 at 0 (pre-treatment), 6 and 24 months. HPV DNA and other related biomarkers status at 6 and 24 months, treatment failures at 24 months and condom use compliance rates represented study outcomes. Results: Six months post-operatively we documented a reduction in the rates of HPV DNA positivity, which was detected in only 23.2% of compliant condom users in comparison to 61.9% in the non-compliant group (*p* < 0.001, OR: 0.19, 95%CI: 0.1–0.36). For the HPV mRNA test, either assessed with the NASBA method or with flow cytometry, reduced positivity percentages were observed in the compliant group, in particular 1.6% vs. 8% for NASBA and 7.1% vs. 16.4% using flow cytometry, although these differences were not statistically significant (*p* = 0.1039 and 0.0791, respectively). Finally, reduced p16 positivity rates were documented in the compliant group. At the two year follow up, a more pronounced difference in HPV DNA positivity rates was observed, specifically only 13% positivity among the compliant women compared with 71% of the non-compliant (*p* < 0.0001); this illustrates a further decreasing trend compared with the 6th month in the compliant group as opposed to an increasing tendency in the non-compliant group, respectively (difference: 9.0%, 95% CI: 0% to 20.6%, *p* = 0.1523). At that time, 80% of the failed treatments were HPV mRNA positive compared to 10% positivity for the cases treated successfully (OR: 34, 95%CI: 6.8–173, *p* < 0.0001), a finding indicative that HPV mRNA E6 & E7 positivity accurately predicts treatment failure; p16 positivity was also observed at higher rates in cases with treatment failure. Conclusions: Consistent condom use following conservative excisional CIN treatment appears to significantly reduce rates of CIN recurrence and biomarkers of HPV expression. Additional HPV vaccination at the time of treatment could further enhance the positive effect of consistent condom use.

## 1. Introduction

The purpose of global cervical screening programs, either cytology or HPV (Human Papillomavirus) biomarker based, focuses on the early detection and treatment of preinvasive lesions and, ultimately, on the reduction of cervical cancer incidence and mortality [1,2,3]. Well-organised systematic call and recall screening programmes with appropriate treatment of screen-detected cervical precancer have resulted in profound decreases in invasive disease incidence (by up to 80%) [4,5,6].

Approximately 10% of the total population screened in the UK will have an abnormal smear; a proportion of those will ultimately require treatment [7,8]. The available conservative outpatient treatment methods, be they ablative or excisional, have both high cure rates of over 90% [9,10,11,12,13]. Despite the high success rates, a fraction of treated women (5–10%) will require repeat treatment for residual or recurrent disease [9,10]. Women post-treatment also remain at 4–5 times greater risk of future invasive disease for at least 2 decades in comparison to the general population, with individuals over 50 years of age being at higher risk [11,14]. Seemingly, no technique that would uniformly achieve optimal disease eradication currently exists [13].

For most women, the exact pathological course leading to treatment failure, pre-invasive or invasive disease following treatment remains unclear. Inadequate technique, residual HPV infection, re-infection with (other) high-risk oncogenic HPV or crypt involvement might all represent the underlying plausible mechanism [15]. Cytology, colposcopy, HPV DNA test and more recently, established or novel HPV related biomarkers have been proposed and used successfully in the post-treatment surveillance period giving excellent results [16,17,18,19,20,21,22]. HPV DNA test, in particular, seems to further improve accuracy and has been established as a ‘test of cure’ (TOC) which allows women to return back to community recall [23]. Despite the numerous tests and biomarkers which have been extensively investigated for the prediction and early detection of treatment failures, effective preventative measures are missing. Vaccination with the existing Virus Like Particle (VLP) vaccine following local treatment represents a promising strategy; however, large randomized control trials (RCTs) on the role of vaccination post conization are pending [24,25].

The majority of women suffering from cervical precancer and cancer are young and commonly desiring future fertility [26]. Evidence in the literature reports marginally higher perinatal morbidity risk after a single excisional procedure (Large Loop Excision of the Transformation Zone -LLETZ), which remains the mainstay modality in cervical precancer treatment; this risk increases exponentially following repeat conizations due to treatment failures (incomplete/inadequate excisions); investigation to identify potential ways to prevent re-excision should be encouraged [27,28,29,30,31,32,33].

Oddly, the role of condom use in the prevention of HPV infections and cervical intraepithelial disease has not been exhaustively investigated so far [34,35]. Data suggests that consistent condom use might reduce HPV infection rates and, as a result, intra-epithelial disease development in HPV-naïve women [36]. Its use also favours lower progression rates to high-grade Cervical Intraepithelial Neoplasia (CIN 2–3) or invasive disease; it also protects against anogenital warts [34]. Evidence suggests that condom use in HPV positive couples may reduce transmission rates between sexual partners as well as viral load, may enhance HPV clearance and might ultimately influence the clinical course of CIN leading to regression [37].

Evidence on the role of condom use after CIN treatment was sparse until fairly recently. The only published data concerning the effect of condom use after excisional treatment for CIN has been illustrated in a previous work of our group, which demonstrated that consistent use of condoms post operatively significantly reduces HPV positivity rates at the 6^th^ post-operative month [38]. Surgical local treatment itself increases HPV negativity rates post treatment by removing the primary lesion, but might not achieve clearance of any possible residual disease; thus the most plausible explanation for the beneficial effects of consistent condom use in this population could be that the “barrier” may prevent re-infection, promote HPV clearance and regression of residual disease and ultimately lead to reduced treatment failure rates [38]. In the “vaccination post conisation” strategy, optimal treatment, HPV clearance and maintenance of HPV naivety till the completion of the full vaccine schedule is even more imperative [39]. However, among the major disadvantages of condoms are the partial protection they confer, while the poor compliance for long term periods between partners represents an important hurdle.

This prospective pragmatic observational study aimed to assess the effect of condom use versus routine post-operative standard care surveillance on HPV DNA status, other HPV-related biomarkers such as High Risk (HR) E6 & E7 mRNA and cytological p16 and ultimately on treatment failure rates. The study was conducted at the colposcopy clinic of the University Hospital of Ioannina from May 2008 to April 2014.

## 2. Materials and Methods

### 2.1. Study Population Inclusion and Exclusion Criteria

We included women of reproductive age who were referred to the University Hospital of Ioannina colposcopy clinic for evaluation of abnormal cytology for whom a decision for excisional treatment for CIN was made. This clinic constitutes the referral centre for North West Greece. At that time the colposcopy clinic adhered to a “see and treat” policy; most cases presenting with high grade cytology as well as several cases of persistent low grade dyskaryosis received excisional treatment in the form of LLETZ.

Besides including all women of reproductive age who signed the informed consent form presenting with any grade of cytological abnormalities and/or abnormal colposcopy (Low Grade Squamous Intraepithelial Lesion–LGSIL/High Grade Squamous Intraepithelial Lesion–HGSIL) for which a decision for excisional treatment by LLETZ was made, we also included individuals who were referred testing positive for HR HPV DNA genotyping at the initial visit whose colposcopic findings warranted excisional treatment (LGSIL/HGSIL).

We excluded women in which histology of the LLETZ specimen revealed invasive disease, involved endocervical margins or had no evidence of CIN, those who opted to undergo hysterectomy, women with sequential excisional treatments, individuals who were pregnant at the time of enrolment as well as women for whom a pre-treatment Liquid Based Cytology (LBC) sample was unavailable. Individuals who declined signing the study’s informed consent form were also excluded from the analysis.

### 2.2. Study Protocol

Based on the study protocol, in all the women, an extensive gynecological history was obtained at the first visit. This detailed specific history addressed age at coitarche, age at first sexual intercourse, lifetime number of sexual partners since coitarche and the use of condoms. In addition to these epidemiological data, other confounding factors affecting HPV and CIN (e.g., smoking) were also recorded. A standard questionnaire implemented in most Hellenic Cervical Pathology Academic (HeCPA) group protocols that incorporated questions on condom use was utilized. Specifically, the questions regarding condom use were focused on the percentage and frequency of condom usage in sexual intercourses as well as the duration of usage in terms of the reason (e.g., contraception) and the status of sexual partner [24].

In all participating individuals an LBC sample was obtained using a Rovers™ Cervex-brush just prior to the colposcopic evaluation. This was transferred in PreservCyt solution and subsequently underwent cytological and bio-molecular analysis for established HPV related biomarkers. All the specimens were centrally analyzed at the Attikon University Hospital of Athens. The cytological examination was expressed according to the Bethesda classification (TBS 2001 system) [40,41].

HPV DNA typing was performed using the CLART^™^ (Clinical Array Technology) Human Papillomavirus 2 kit that detects 35 different HPV genotypes (high or low risk) by PCR amplification of a fragment within the highly conserved L1 region of the virus [42]. The polymerase chain reaction (PCR) amplification technique for specific fragments of the viral genome and their hybridization with specific probes for each HPV type was used to detect infections and coinfections of the following High Risk-HPV types: 16, 18, 26, 31, 33, 35, 39, 43, 45, 51, 52, 53, 56, 58, 59, 66, 68, 70, 73, 82 and 85 and the Low-Risk HPV types 6, 11, 40, 42, 44, 54, 61, 62, 71, 72, 81, 83, 84 and 89.

Nucleic Acid Sequence Based Amplification (NASBA) and multiplex detection assays (NucliSENS EasyQ HPV v1.0™), a real-time (nucleic acid) sequence-based assay, were used for the qualitative determination of E6/E7 mRNAs of the five most commonly identified carcinogenic HPV genotypes (HPV 16, 18, 31, 33 and 45) [43].

Flow cytometric evaluation of E6/E7 mRNA of high-risk HPV types (16, 18, 31, 33, 35, 39, 45, 51, 52, 56, 58, 59, 68, 73 and 82) was performed with HPV OncoTect™ (Invirion Diagnostics, Oakbrook, IL, USA) [43].

Finally, p16 immunostaining p16INK4a was performed using the CINtec™ Cytology Kit (ROCHE, Basel, Switzerland). Staining for p16 was considered positive if at least one dysplastic cell was stained for the marker [44]. Furthermore, the Dako AutoStainer system was used for the staining of the smears according to the standard protocol for Thinprep^®^ samples.

Based on the study design, all women underwent a colposcopic evaluation in order to document possible cytologic and biomolecular discrepancies. All these examinations were performed by expert board-accredited colposcopists. After the initial colposcopic evaluation, women for whom a decision for excisional treatment was made were scheduled for a LLETZ procedure shortly after their menstrual period [45]. Following treatment, all individuals received standard postoperative consultation including a strong recommendation for consistent condom use until the 1st follow up visit (6th month post-operatively). At the 6th month visit a second LBC sample was obtained; this was sent for cytological and biomolecular analysis of the same HPV-related biomarkers just prior to follow up colposcopic evaluation. All these data were recorded and women were asked to complete the standard questionnaire regarding condom use. The same assessment (LBC, biomarkers analysis, colposcopy and standard questionnaire completion was repeated in the 24-month follow up visit. All individuals in which recurrent disease was documented were classified as treatment failures; for these women a second LLETZ procedure was planned. Finally, along with the interpretation of the biomolecular and clinical results, we analyzed the questionnaire answers regarding percentage of condom use.

In all HPV unvaccinated individuals, a strong recommendation for HPV vaccination was given irrespective of the particular cytological and or colposcopical findings, since this represents universal standard clinical policy of this department.

All women were informed about the scope of the study and were asked to sign a consent form before entering the study. The study’s protocol has been approved by the Ioannina University Hospital’s ethical committee [protocol 28/9-7-2009(Θ.21)] as well as the Greek Central Government (Ministry of Education and Religious Affairs), under the frame of the HPVGuard research project (http://HPVGuard.org, Project Number: 11ΣΥΝ_10_250, Cooperation framework, Protocol Number: ΕΥΔΕ–ΕΤAΚ 1788/1-10-2012), and subsequently received additional approval from the coordinating authority “Attikon” University Hospital Ethics Committee (Code: ΕΒΔ 623/14-5-13) [46].

### 2.3. Statistical Analysis

The statistical analysis was performed with the SAS for Windows 9.4 statistical analysis software (SAS Institute Inc., Cary, NC, USA). Descriptive values are expressed as median and quartile 1 to quartile 3 range (Q1–Q3) while for the categorical data the frequency and the relevant percentages are presented. Comparisons between groups for the qualitative parameters was made using the chi-square test (and if required the Fisher exact test). For the arithmetic parameters (such as woman’s age, age of first sexual intercourse or number of lifetime sex partners) normality could not be ensured; therefore, non-parametric tests were applied, specifically the Kruskal–Wallis test. The significance level (*p*-value) was set to 0.05 and all tests were two sided. Power calculation (sample size) could not be calculated at the time of the initial study design; however, we performed a post hoc analysis to estimate the power of the obtained results after data collection. This power analysis focused on treatment failures (10 cases) compared to successful treatments (194 cases). Power analysis was performed using G*Power software version 3.1.9.6, implementing the Exact test family; for proportions of two independent groups, error probability was set as α = 0.05 and the tests were assumed two tailed. Power of the tests was reported as 1-β.

## 3. Results

### 3.1. Demographic Data

In total, 204 women successfully completed the two-year follow-up and a full HPV biomarker dataset was available, both for enrolment, the six months assessment as well as the two-year follow-up visit. All women received a strong recommendation for consistent condom use for the first 6 months post-op. In terms of compliancy, 34.4% (70 individuals) of the women responded that they had consistently used condoms during intercourse (more than 90% of instances) throughout the first 6 months post-op period, a higher percentage (47.5%) showed minimal adherence (<10% of instances), 16 women declared condom use in 25% of their sexual activities, 10 women (4.9%) in 50% of intercourse; finally 11 women (5.4%) reported acceptable rates of condom use (75% of instances).

The group of 70 women who consistently (i.e., ≥90% of sexual intercourses) used condoms was compared with the remaining women which occasionally used them (at lower percentages: N = 134).

The baseline characteristics of the two groups (≥90% condom use vs. <90% condom use) are presented in Table 1. Compliant and non-compliant women did not present important differences in their demographic and sexual behavior characteristics, or on their reproductive or colposcopic characteristics as these were assessed by referral cytology and colposcopic impression upon study entry.

Detailed information for the histological outcomes and consistency with colposcopic impression along with HR HPV DNA percentages for each group are presented in Table 2.

### 3.2. Biomarkers’ Data

At baseline (i.e., before treatment) the two groups had similar HPV biomarker profiles (see Table 3), specifically 87% and 85% of the compliant and non-compliant women were HPV DNA positive (*p* = 0.6880) and the results for HPV mRNA E6 & E7 positivity were 45% and 49% respectively (*p* = 0.6075). Flow cytometry and p16 results were also similar between the two groups (*p* > 0.05 for both comparisons, see Table 3).

Six months post treatment we documented a reduction in the expression of HPV-related biomarkers in the group that used condoms consistently. Specifically, only 23.2% of consistent condom users (≥90% of instances) tested positive for HPV DNA compared to 61.9% in the group of inconsistent users (<90%); (*p* < 0.001, OR: 0.19 95%CI: 0.1–0.36). As for HPV mRNA E6 & E7 expression, whether tested with the NASBA method or with flow cytometry, reduced positivity rates were documented in the compliant group; 1.6% vs. 8% for NASBA and 7.1% vs. 16.4% for flow cytometry, but without reaching statistical significance (*p* = 0.1039 and 0.0791, respectively). Finally, p16 positivity rates were favorable in the compliant group (see detailed Table 4).

At the two year follow up, although women were not advised to continue using condoms (however, they were not discouraged from doing so), an increased difference of HPV DNA positivity rates was observed. In particular, 13% of the compliant women tested positive whereas positivity reached 71% in the non-compliant group (*p* < 0.0001). Moreover, the positivity rate dropped from 23.2% at the 6th month in the compliant group to 13% at the second-year assessment (difference: 10.2%, 95% CI 0% to 23.79%, *p* = 0.1819). In contrast, for the non-compliant group the positivity rate increased from 62% at the 6th month to 71% at the second-year (difference: 9.0%, 95% CI: 0% to 20.6%, *p* = 0.1523) (see Table 5).

From a statistical point of view, although the number of cases with treatment failures was rather small compared to non-failures (N = 194) we applied various tests to identify possible prognostic factors. As for the ten women with treatment failures (4.9%), one belonged to the compliant group (failure percentage: 1.4%), while 9 were in the non-compliant group (failure percentage: 6.7%); obviously the small number of treatment failures poses difficulties in terms of comparisons (*p* = 0.1693, OR: 0.20, 95% CI: 0.03–1.62, 1-β = 15.8%). We could arbitrarily postulate that in a larger setting, compliant women with consistent condom use would have 5 times less odds for failure. Parity and mode of delivery (i.e., vaginal delivery versus cesarean section) did not represent significant factors predicting treatment failure. Among the studied biomarkers, HPV DNA test results were significantly different; all failed treatments were HPV DNA positive while from the successful treatments 48.7% were positive (*p* = 0.0016, RR:0.49, 95% CI: 0.42–0.56, 1-β = 99.9%). All relapsed cases were also HPV DNA positive at enrollment; however, three of them were negative when tested at 6 months. HPV mRNA E6 & E7 positivity rate was also increased in cases with treatment failure; 40% of the relapsed cases were positive at 6 months compared to 4% of the cases treated successfully (OR: 16, 95%CI: 3.7–70, 1-β = 85.7%). At the 24 months check point, 80% of the failed treatments were HPV mRNA E6 & E7 positive compared to 10% positivity for the cases treated successfully (OR: 34, 95%CI: 6.8–173, *p* < 0.0001, 1-β = 99.9%). This was also evident in flow cytometry results at 6 and 24 months (*p* = 0.1288, OR: 3.07, 95% CI: 0.74–12.73, 1-β = 31.8% and *p* = 0.0011, OR: 9.96, 95%CI: 2.63–37.78, 1-β = 90.9% respectively), a finding indicative that HPV mRNA E6 & E7 positivity accurately predicts treatment failure. Finally, p16 positivity was observed at higher rates in cases with treatment failure as compared with those which were treated successfully, corroborating this biomarker’s high predictive value (see Table 6). A multivariable analysis might be informative; however, from a statistical standpoint a larger sample would be required.

## 4. Discussion

Even following adequate local treatment, women with previous CIN remain at an approximately 4 to 5-fold elevated risk of subsequently developing cervical cancer for the following 25 years, or even their entire lifespan. This principle was first conceptualized by Soutter et al. and has been repeatedly corroborated since by numerous observational studies and well-designed randomized trials worldwide [11,14,47]. Based on the new cervical cancer prevention paradigm, HPV biomarker-based screening and TOC offer superior negative predictive value and reassurance, allowing prolonged screening intervals. Concerns do exist, however, as to whether it is safe for individuals with previous surgery for cervical precancer to undergo screening at these prolonged intervals.

The relative contribution of factors leading to recurrent or residual disease post conization still represents an open case [48,49]. Some authors advocate that recurrent CIN can be accurately predicted at the time of conization itself; this annuls the potential effect of cofactors promoting disease relapse during the healing process [50]. Despite the extensive research so far, the role of condom use in the natural history of HPV infection still remains controversial with substantial heterogeneity between studies. A hypothetical, yet unproved, explanation of condoms beneficial effect might rely on the disruption of continuous transmission of shed HPV particles between sexual partners [37]. In an older meta-analysis critical appraisal, Manhart and Koutsky advocate that while condoms may not prevent HPV infection itself, they may protect against genital warts, HGSIL and invasive cervical cancer [34]. The prospective study of Winer et al. demonstrated an inverse, temporal association between the frequency of condom use by male partners and the risk of HPV infection in women [36]. This association was strong and increased with the increasing frequency of condom use, suggesting a causal, protective effect.

In a pooled analysis of the older International Agency for Research in Cancer (IARC) studies, no protective effect of condom use has been established; the authors do, however, acknowledge that a possible beneficial effect has been perhaps underestimated, as very few women used condoms consistently throughout their lives [51]. Several other confounding factors need addressing; for instance, in the study of de Sanjose et al., monogamous women reported use of condoms for longer periods as compared with non-monogamous women, highlighting that the protection conferred by condom use might reflect both the barrier effect against HPV infection as well as a pattern of sexual behavior less likely to promote exposure to HPV [52].

In a more recent systematic review of eight longitudinal studies, Lam et al. conclude that consistent condom use appears to offer a relatively good protection from HPV infections and associated cervical neoplasia. The authors correctly identify women with previous surgical treatment of CIN as another group who could benefit from “consistent and correct condom use” [53].

Our group pioneered study of the contributing factors for persistent HPV-biomarker positivity following conservative surgical treatment of CIN [38]. In this older follow up study, the multivariate analysis demonstrated that consistent condom use post-treatment significantly reduced the short term HR-HPV positivity rates in comparison to no use. Thus, consistent condom use for the first six months post treatment emerged as the main predictive factor for disease relapse. We had then proposed that consistent use potentially either prevents a new HPV infection by a novel genotype, or re-infection by the same HPV type from the sexual partner, or even facilitates clearance of any residual HPV infection. The latter hypothesis is supported by the observation that in both studies condom use was associated with a significant reduction in mRNA and p16 positivity rates post-operatively.

Based on its high sensitivity, detection of HR-HPV DNA represents a long-established diagnostic TOC following CIN treatment. Admittedly, the universal adoption of this biomarker-based surveillance policy caused significant increases in colposcopy referrals, while assay choice might apparently impact this burden [54,55]. The case of concurrent HPV genotyping poses two major advantages by: (i) characterizing type-specific persistence post conization, which might be related to elevated risk for subsequent CIN2+ development, as well as (ii) identifying HPV-16 persistence which is linked with significantly increased risk of residual or recurrent disease compared to other HR-HPV types [55]. In our study, consistent condom users following treatment illustrated significantly lower probability of testing HPV positive at 6 and 24 post-op months besides having subsequently lower chances for treatment failure. Specifically, six months following recruitment, women reporting consistent condom use exhibited a 23% HPV DNA positivity rate, while women with inconsistent use presented an almost three-fold higher rate (62%); thus, a ≥90% use of condoms decreased the odds for HPV positivity (OR: 0.19, 95%CI: 0.10–0.36, *p* < 0.0001). Two years later, women with consistent condom use had even lower HPV DNA positivity rates (13%) while 71% of women with inconsistent use tested HPV DNA positive (OR: 0.06, 95%CI: 0.03–0.13, *p* < 0.000). In summary, women with consistent condom use not only had lower HPV DNA positivity rates both after 6 and 24 months, as compared to women with inconsistent condom use, but, additionally, they illustrated a decreasing trend for HPV DNA positivity rate over time (from 23% to 13%), while inconsistent users had an increasing HPV positivity rate (from 62% to 71%).

Despite its inherent inability to detect treatment failures attributed to intermediate risk HPV’s, the role of mRNA HPV as a credible TOC also emerges in the literature. In a sample of 116 women, Zappacosta et al. explored the possible role of mRNA HPV assessment by NASBA on the detection of residual/recurrent cervical disease after successful LLETZ [56]. These authors concluded that the mRNA test showed higher specificity and positive predictive value than the combination cytology-plus-HPV-DNA test. They considered the detection of HPV oncogenic mRNA transcripts as the best indicator of the risk of developing CIN [56]. Several years later, Tisi et al. also studied in a smaller sample of 43 women the role of HPV DNA, HPV mRNA and cytology in the follow-up of women treated for cervical dysplasia [57]. This group also considered that HPV mRNA test has higher specificity with respect to cytology and HPV DNA in a TOC setting, avoiding the referral to unnecessary colposcopy with benefits for healthcare systems [57]. In our study, at 6-months post-op, although the percentage of HPV mRNA positive women was lower in the consistent users’ group (1.6%/7.1%, NASBA/flow outcomes) than in the inconsistent users’ group (8%/16.4% ,NASBA/flow results), this did not reach statistical significance, due to the small number of positive cases. In contrast, at 24 months assessment, higher mRNA E6 & E7 positivity rates were documented in both groups (6.3%/10% NASBA/flow outcomes, and 17.8%/18.3% NASBA/flow results), a finding marginally significant (*p* = 0.0448) when considering the NASBA method but insignificant for the flow approach. Nevertheless, HPV mRNA positivity rates for compliant women were again lower than incompliant ones, despite exhibiting an increasing trend longitudinally for both groups, a finding requiring further investigation.

Treatment failures at 24 months post treatment reflect the cumulative effect of several contributing factors which may act synergistically throughout a prolonged period: inconsistent condom use, sequential sexual relationships, casual partners as well as partner concurrency (overlapping sexual partnerships). Furthermore, unopposed estrogenic stimulation from persistent ovarian cysts, cervical pathogens *(Ureaplasma* sp., *Chlamydia* sp. etc.), other STI’s, alterations in vaginal metabolome proteomics and epigenomics might also interfere with the healing process [58,59]. Even for consistent condom use, the relative importance of all these confounding factors is difficult to quantify.

From a clinical standpoint, the conization procedure itself theoretically removes the affected site and clears the “nest” of HPV infection (entirely or almost so). Subsequent consistent condom use could prevent the re-inoculation of the shed HPV particles in the rapidly transforming conization crater. Another plausible scenario is that the excisional treatment itself and ball cauterization remove and destroy accordingly the lesion and pathological tissue predominantly carrying the viral load. Thus, the diminished residual viral load in addition to subsequent condom use minimizes the HPV re-infection rates, giving a time window for the development of systemic immunity and production of antibodies against the causal HPV genotypes. By the end of the six-month period in most patients the healing process has been completed; then condoms can still be occasionally used. Obviously, consistent condom use helps avert numerous other pathogens that interfere with the healing process, or sexually transmitted infections acting synergistically with HR-HPVs such as *Chlamydia* sp.

Of note is that most women undergoing conservative cervical surgery would be universally counselled to avoid future pregnancies whatsoever for a 12-month period during which not only the healing process has been completed, but also cervical competence has been optimized [32,38]. This evidence-based advice for the postoperative period is currently adopted by most departments worldwide. From this perspective, consistent condom use will serve both purposes (family planning as well as protection against HPV) [25,35,38].

As we were unable to identify in the literature previous studies addressing the effects of consistent condom use on HPV biomarker’s expression in women following CIN treatment, the main strength of this study relies on its contribution to a field with limited evidence. The availability of prolonged follow-up data represents the core value; despite the late assessment, 24 months after the intervention admittedly reflects the sum effect of several contributing factors besides condom use, as already analyzed.

We are also cognizant of our study’s main limitations. Both mRNA assays utilized in this study are now considered outdated as they are rarely being used in current clinical practice; however, when assessed in combination they qualitatively do approach the strengths of the APTIMA assay [Hologic, Marlborough, Massachusetts, etc.] which is currently mostly implemented. Another important shortfall of this study can be tracked down to its initial conception when no stratification based on HPV-vaccination status was foreseen. HPV vaccination rates were low at the time the study started to recruit; there was also no evidence of the potential role of vaccination following conization in preventing recurrence [25,60]. Of note is that our group has previously illustrated a beneficial contribution of HPV vaccination affecting untreated patients with mild dysplasias under colposcopic surveillance [25,35].

## 5. Conclusions

In this study we have shown that consistent condom use can increase HPV biomarker negativity rates not only in the short term but also at 24 months post-operatively; thus, it could reduce treatment failure rates expressed as either residual or recurrent disease as well as, in the long term, the risk of invasive cervical cancer. If corroborated in a larger setting, then we could hypothesize that compliant women with consistent condom use would have five times lower odds for treatment failure. Aiming to secure and prolong this protective effect on a realistic basis, anti-HPV vaccination with the current VLP vaccines could be suggested. The virtues of vaccine recommendation for non–treated patients harboring cervical precancer have been also corroborated in a recent Greek multicenter observational cohort study of the HeCPA group [24,25].

The favorable effect of consistent condom use following conization is inherently related to the concept of residual and recurrent cervical disease and ultimately to uncertainties on HPV latency, reflecting gaps in knowledge of the HPV natural cycle. With almost 30% of incident detections being attributed to likely re-detection of prior infection, the evidence in the literature is that a ‘cleared’ HPV result may truly indicate viral eradication or may represent control of the infection below the limits of detection, also referred to as ‘HPV latency’ [61,62]. Data from the mid-adult female vaccine trials favor vaccine effectiveness also in women who have baseline antibodies against vaccine type HPV, indicating past infection. The authors consider that both the HPV-FASTER concept, as well as other initiatives seeking to implement an integrated screening and vaccination program in women up to age 45 years, could benefit from evidence from studies designed to address vaccine effectiveness in women with prior HPV infection [63]. Insight on HPV latency would help clinicians in their counseling regarding patients’ concerns about their HR-HPV test results. From a public health perspective, accurate natural history models are necessary for updating evidence-based HPV vaccination and cervical cancer screening strategies for older populations [61,62].

An equally ambitious yet feasible project could be the further validation of a Cervical Pathology Scoring system, a concept which was initially introduced in published form by Paraskevaidis et al., several years before the introduction of commercially available colposcopy mobile apps [17,18,20,33]. Together with information such as vaccination status, age at onset of sexual intercourse, number of sexual partners and smoking habits, frequency of condom use represents an integral part of a Lifestyle Cervical Pathology Risk Assessment Tool during history taking at the colposcopy clinic [19,24,35]. When validated and combined with cytology and HPV-biomarker data, this user-friendly algorithm would represent an invaluable tool for the clinician, predicting accurately the likely clinical course of every new colposcopy referral. Novel approaches, such as self-sampling and first void urine HPV in individuals non-compliant with regular post-operative visits, could be proposed in terms of long term follow up to detect recurrence disease and treatment failures [64].

## Figures and Tables

**Table 1 cancers-14-03530-t001:** Baseline characteristics of the study population.

	Consistent Use (≥90%) (N = 70)	Inconsistent Use (<90%) (N = 134)	*p*
Age (Median, Q1–Q3)	37.5 (33–43)	38 (34–41)	0.8995
Number of children (Median, Q1–Q3)	1 (0–2)	1 (0–2)	0.2803
Age of sexual activity initiation (Median, Q1–Q3)	19 (18–20)	19 (17–20)	0.5665
Number of sex partners (Median, Q1–Q3)	3 (2–5)	3 (2–5)	0.7960
Smoking in packet years (Median, Q1–Q3)	2 (0–10)	3 (0–10)	0.8719
Parity (N, %)	45 (64.3%)	92 (68.7%)	0.5280
Delivery via CS (N, %)	16 (35.6%)	37 (40.2)	0.5988
Referral cytology			0.1448
NILM	0 (0%)	1 (0.75%)	
HPV	6 (8.57%)	15 (11.19%)	
LGSIL	28 (40%)	31 (23.13%)	
ASC-US	5 (7.14%)	14 (10.45%)	
ASC-H	0 (0%)	6 (4.48%)	
AGC	1 (1.43%)	3 (2.24%)	
HGSIL	30 (42.86%)	64 (47.76%)	
Colposcopy on study entry (N, %)			0.6189
Negative	0 (0%)	1 (0.75%)	
HPV	4 (5.71%)	6 (4.48%)	
LGSIL	30 (42.86%)	47 (35.07%)	
HGSIL	36 (51.43%)	80 (59.7%)	

Q1: 1st quartile, Q3: 3rd quartile, CS: Cesarean Section, NILM: Negative for Intraepithelial Lesion or Malignancy, LGSIL: Low Grade Squamous Intraepithelial neoplasia, ASC-US: Atypical Squamous Cells of Undetermined Significance, ASC-H: Atypical Squamous Cells with High probability for Malignancy, AGC: Atypical Glandular Cells: HGSIL: High Grade Squamous Intraepithelial neoplasia, HPV: Human Papilloma Virus. N: Number of cases.

**Table 2 cancers-14-03530-t002:** Histological results and consistency with colposcopic findings at study entry.

	Histology					Cumulative HR HPV DNA Positivity
Colposcopic Impression	No CIN evidence	CIN-1	CIN-2	CIN-3	Micro invasion	
Negative	-	-	1 (100%)	-	-	1 (100%)
HPV	-	9 (78%)	1 (100%)	-	-	10 (80%)
LGSIL	-	63 (43%)	13 (92%)	1 (100%)	-	77 (52%)
HGSIL	-	9 (56%)	42 (90%)	63 (98%)	2 (100%)	116 (92%)
Total	-	81 (48%)	57 (91%)	64 (98%)	2 (100%)	204 (76%)

Within parentheses is the percentage of HR HPV DNA positive cases for each subgroup. The CIN-3 histological category also includes the only two cases with glandular histology (cGIN).

**Table 3 cancers-14-03530-t003:** Laboratory outcomes for the two groups and statistical comparison during the study entry.

			Consistent Use (≥90%) (N = 70)		Inconsistent Use (<90%) (N = 134)			
	Consistent (N)	Inconsistent (N)	Positives	Negatives	Positives	Negatives	*p*	OR (95% CI)
HPV DNA test results	70	134	61 (87.1%)	9 (12.9%)	114 (85.1%)	20 (14.9%)	0.6880	1.2 (0.5–2.8)
NASBA results	64	130	29 (45.3%)	35 (54.7%)	64 (49.2%)	66 (50.8%)	0.6075	0.85 (0.47–1.6)
Flow cytometry results	70	133	40 (57.1%)	30 (42.9%)	72 (54.1%)	61 (45.9%)	0.6821	1.1 (0.6–2.0)
p16 results	67	132	22 (32.8%)	45 (67.2%)	41 (31.1%)	91 (68.9%)	0.7992	1.1 (0.6–2.0)

**Table 4 cancers-14-03530-t004:** Laboratory outcomes for the two groups and statistical comparison at the 6 months follow up.

			Consistent Use (≥90%) (N = 70)		Inconsistent Use (<90%) (N = 134)			
	Consistent (N)	Inconsistent (N)	Positives	Negatives	Positives	Negatives	*p*	OR/RR (95% CI)
HPV DNA test results	69	134	16 (23.2%)	53 (76.8%)	83 (61.9%)	51 (38.1%)	<0.0001	0.19 (0.10–0.36)
NASBA results	62	125	1 (1.6%)	61 (98.4%)	10 (8%)	115 (92%)	0.1039	0.19 (0.02–1.51)
Flow cytometry results	70	128	5 (7.1%)	65 (92.9%)	21 (16.4%)	107 (83.6%)	0.0791	0.39 (0.14–1.09)
p16 results	67	125	0 (0%)	67 (100%)	6 (4.8%)	119 (95.2%)	0.0983	RR: 0.64 (0.57–0.71)

**Table 5 cancers-14-03530-t005:** Laboratory outcomes for the two groups and statistical comparison after 24 months of follow up.

			Consistent Use (≥90%) (N = 70)		Inconsistent Use (<90%) (N = 134)			
	Consistent (N)	Inconsistent (N)	Positives	Negatives	Positives	Negatives	*p*	OR (95% CI)
HPV DNA test results	69	134	9 (13%)	60 (87%)	95 (70.9%)	39 (29.1%)	<0.0001	0.06 (0.03–0.13)
NASBA results	63	129	4 (6.3%)	59 (93.7%)	23 (17.8%)	106 (82.2%)	0.0448	0.31 (0.10–0.95)
Flow cytometry results	70	131	7 (10%)	63 (90%)	24 (18.3%)	107 (81.7%)	0.1521	0.50 (0.20–1.22)
p16 results	63	121	1 (1.6%)	62 (98.4%)	6 (5%)	115 (95%)	0.1686	0.23 (0.03–1.86)

**Table 6 cancers-14-03530-t006:** Results from statistical comparisons between treatment failures and successful treatments.

Description	Treatment Failures (N = 10)	Successful Treatments (N = 194)	*p*	OR/RR and 95% CI
HPV DNA test results at enrollment (positive)	10/100%	165/85.05%	0.3631	RR: 0.85 (0.80–0.90)
HPV DNA test results at 6 months (positive)	7/70%	92/47.67%	0.2054	2.56 (0.64–10.2)
HPV DNA test results at 24 months (positive)	10/100%	94/48.7%	0.0016	RR: 0.49 (0.42–0.56)
HPV mRNA test results at enrollment (positive)	6/60%	87/47.28%	0.5244	1.67 (0.46–6.12)
HPV mRNA test results at 6 months (positive)	4/40%	7/3.96%	0.0012	16.19 (3.71–70.68)
HPV mRNA test results at 24 months (positive)	8/80%	19/10.44%	<0.0001	34.32 (6.79–173.52)
Flow cytometry at enrollment (positive)	7/70%	105/54.4%	0.5168	1.96 (0.49–7.79)
Flow cytometry at 6 months (positive)	3/30%	23/12.23%	0.1288	3.07 (0.74–12.73)
Flow cytometry at 24 months (positive)	6/60%	25/13.09%	0.0011	9.96 (2.63–37.78)
p16 at enrollment (positive)	7/70%	56/29.63%	0.0125	5.54 (1.38–22.21)
p16 at 6 months (positive)	3/30%	4/2.2%	0.0032	19.07 (3.57–101.99)
p16 at 24 months (positive)	6/60%	3/1.72%	<0.0001	85.5 (15.56–469.89)
Condom use (compliant >90%)	1/10%	69/35.57%	0.1693	0.2 (0.03–1.62)
Parity (Yes)	9/90%	128/65.98%	0.1707	4.64 (0.58–37.42)
Mode of delivery (VD)	6/66.7%	78/60.94%	1.0000	1.28 (0.31–5.36)

CS: Caesarean Section VD: Vaginal Delivery, OR: Odds Ratio, RR: Relative Risk, N: Number of cases

## Data Availability

Data are available upon request from the corresponding author.

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
