# Peer review of "Effect of Condom Use after CIN Treatment on Cervical HPV Biomarkers Positivity: Prolonged Follow Up Study"

_cancers, 2022, doi:10.3390/cancers14143530_

Round 1
Reviewer 1 Report
Dear Authors
Thank you for the opportunity to get know with your interesting research
The study comprises an important question of decreasing the risk for recurrence of CIN2/3 after excisional procedure associated with reinfection or persistence of high risk HPV.
For the practical utility, results of the study should be presented, however there are some doubts in methodology which should be claryfied.
1. Please present the clear indications for LLETZ procedure
2. Please reveal the histological results for all patients - consistence with colposcopic findings and percentage of high risk changes - I could not find any specification what kind of specific lesions were diagnosed (CIN1, 2/3)
3. The limitation of the study is the number of cases included. 204 of all cases and ten cases of recurrence make the doubts about enough statistical power for the tests that were used. Inclusion of tests power analysis to support your conclusion is highly recommended.
I will be happy to see you corrected manuscript
Author Response
Reviewer 1
Comment: 1. Thank you for the opportunity to get know with your interesting research.
The study comprises an important question of decreasing the risk for recurrence of CIN2/3 after excisional procedure associated with reinfection or persistence of high risk HPV.
For the practical utility, results of the study should be presented, however there are some doubts in methodology which should be clarified.
Authors’ actions: No action, thank you for your comments, we hope that you find the revised version improved.
Comment: 2. Please present the clear indications for LLETZ procedure.
Authors’ actions: As described in the submitted manuscript (Section 2.1), three were the main indications for a LLETZ procedure at the time of the study in our institution:
- Women presenting with high grade cytology (Some individuals with equivocal colposcopic impression underwent cervical biopsies beforehand).
- Women with persistent low grade dyskaryosis desiring define treatment.
- Referrals testing positive for HR HPV DNA genotyping, whose initial colposcopic findings warranted excisional treatment.
Comment: 3. Please reveal the histological results for all patients - consistence with colposcopic findings and percentage of high risk changes - I could not find any specification what kind of specific lesions were diagnosed (CIN1, 2/3)
Authors’ actions: Thank you for your point; an additional table (Table 2) and a paragraph have been added in the text following table 1.
“Detailed information for the histological outcomes and consistency with colposcopic impression along with HR HPV DNA percentages for each group are presented in table 2.”
Table 2. Histological results and consistency with colposcopic findings at study entry.
|
Histology |
Cumulative HR HPV DNA Positivity |
||||
Colposcopic Impression |
No CIN evidence |
CIN-1 |
CIN-2 |
CIN-3 |
Micro invasion |
|
Negative |
- |
- |
1 (100%) |
- |
- |
1 (100%) |
HPV |
- |
9 (78%) |
1 (100%) |
- |
- |
10 (80%) |
LGSIL |
- |
63 (43%) |
13 (92%) |
1 (100%) |
- |
77 (52%) |
HGSIL |
- |
9 (56%) |
42 (90%) |
63 (98%) |
2 (100%) |
116 (92%) |
Total |
- |
81 (48%) |
57 (91%) |
64 (98%) |
2 (100%) |
204 6%) |
* Within parentheses is the percentage of HR HPV DNA positive cases for each subgroup. In the CIN-3 histological category we have also included the only two cases with glandular histology (cGIN).
Comment: 4. The limitation of the study is the number of cases included. 204 of all cases and ten cases of recurrence make the doubts about enough statistical power for the tests that were used. Inclusion of tests power analysis to support your conclusion is highly recommended.
Authors’ actions: Power analysis specifically for the results related to treatment failures has been added in the revised version. In the section “2.3. Statistical analysis” more details have been added to present the applied methodology and tools for power analysis; additionally, in the results section the power (1-β) for each test related to treatment failures has been also added. Please refer to the revised version with tracked changes.
Reviewer 2 Report
Valasoulis et.al., presented their work in the article, "Effect of condom use after CIN treatment on cervical HPV biomarkers positivity: Prolonged follow up study" where they evaluated the use of the condom after Cervical Intraepithelial Neoplasia (CIN) treatment of HPV positive cervices. There was much enthusiasm to read the article, however there were several limitations in the content of the manuscript. It was hard to go through the entire article due to the very casual utilization of abbreviations through the text. The meaning of CIN for example was not clear or spelled out until the third page of the article. In addition, there were several places where the excessive use of abbreviations, without a full explanation, renders the entire story a bit difficult to comprehend. It is therefore advised that the authors try and describe the abbreviations, conceptually, and in regards to the main message and re-submit the article. Except for this important caveat, the manuscript allows an interesting conclusion on the use of condoms as to the infectivity with HPV post surgery.
Author Response
Reviewer 2
Comment: 1. Valasoulis et.al., presented their work in the article, "Effect of condom use after CIN treatment on cervical HPV biomarkers positivity: Prolonged follow up study" where they evaluated the use of the condom after Cervical Intraepithelial Neoplasia (CIN) treatment of HPV positive cervices. There was much enthusiasm to read the article, however there were several limitations in the content of the manuscript.
Authors’ actions: Thank you for your interest and comments, we hope that in the revised version most limitations have been addressed.
Comment: 2. It was hard to go through the entire article due to the very casual utilization of abbreviations through the text. The meaning of CIN for example was not clear or spelled out until the third page of the article. In addition, there were several places where the excessive use of abbreviations, without a full explanation, renders the entire story a bit difficult to comprehend. It is therefore advised that the authors try and describe the abbreviations, conceptually, and in regards to the main message and re-submit the article. Except for this important caveat, the manuscript allows an interesting conclusion on the use of condoms as to the infectivity with HPV post surgery.
Authors’ actions: Thank you for highlighting an important drawback, in the revised version every effort has been made to simplify the abbreviations and define them upon their first occurrence in the text thus promoting the readability of the manuscript.